# SOLVING NP-HARD PROBLEMS ON GRAPHS WITH EXTENDED ALPHAGO ZERO

## ABSTRACT

There have been increasing challenges to solve combinatorial optimization problems by machine learning. Khalil et al. (NeurIPS 2017) proposed an end-to-end reinforcement learning framework, which automatically learns graph embeddings to construct solutions to a wide range of problems. However, it sometimes performs poorly on graphs having different characteristics than training graphs. To improve its generalization ability to various graphs, we propose a novel learning strategy based on AlphaGo Zero, a Go engine that achieved a superhuman level without the domain knowledge of the game. We redesign AlphaGo Zero for combinatorial optimization problems, taking into account several differences from two-player games. In experiments on five NP-hard problems such as MINIMUMVERTEX-COVER and MAXCUT, our method, with only a policy network, shows better generalization than the previous method to various instances that are not used for training, including random graphs, synthetic graphs, and real-world graphs. Furthermore, our method is significantly enhanced by a test-time Monte Carlo Tree Search which makes full use of the policy network and value network. We also compare recently-developed graph neural network (GNN) models, with an interesting insight into a suitable choice of GNN models for each task.

## 1 INTRODUCTION

There is no polynomial-time algorithm found for NP-hard problems [7], but they often arise in many real-world optimization tasks. Therefore, a variety of algorithms have been developed in a long history, including approximation algorithms [2, 14], meta-heuristics based on local searches such as simulated annealing and evolutionary computation [15, 10], general-purpose exact solvers such as CPLEX [1] and Gurobi [16], and problem-specific exact solvers [1, 25].

Recently, machine learning approaches have been actively investigated to solve combinatorial optimization, with the expectation that the combinatorial structure of the problem can be automatically learned without complicated hand-crafted heuristics. In the early stage, many of these approaches focused on solving specific problems [17, 5] such as the traveling salesperson problem (TSP). Khalil et al. [19] proposed a general framework to solve combinatorial problems by a combination of reinforcement learning and graph embedding, which attracted attention for the following two reasons: It does not require any knowledge on graph algorithms other than greedy selection based on network outputs. Furthermore, it learns algorithms without any training dataset. Thanks to these advantages, the framework can be applied to a diverse range of problems over graphs and it also performs much better than previous learning-based approaches. However, we observed poor empirical performance on some graphs having different characteristics (e.g., synthetic graphs and real-world graphs) than random graphs that were used for training, possibly because of the limited exploration space of their Q-learning method.

In this paper, to overcome its weakness, we propose a novel solver, named CombOpt Zero. CombOpt Zero is inspired by AlphaGo Zero [33], a superhuman engine of Go, which conducts Monte Carlo Tree Search (MCTS) to train deep neural networks. AlphaGo Zero was later generalized to AlphaZero [34] so that it can handle other games; however, its range of applications is limited to two-player games whose state is win/lose (or possibly draw). We extend AlphaGo Zero to a bunch of combinatorial

---

[1] www.cplex.com

problems by a simple normalization technique based on random sampling. In the same way as AlphaGo Zero, CombOpt Zero automatically learns a policy network and value network by self-play based on MCTS. We train our networks for five kinds of NP-hard tasks and test on different instances including standard random graphs (e.g., the Erdős-Renyi model [11] and the Barabási-Albert model [3]), benchmark graphs, and real-world graphs. We show that, with only a greedy selection on the policy network, CombOpt Zero has a better generalization to a variety of graphs than the existing method, which indicates that the MCTS-based training strengthens the exploration of various actions. When more computation time is allowed, using the MCTS at test time with the full use of both the policy network and value network significantly improves the performance. Furthermore, we combine our framework with several graph neural network models [21, 39, 28], and experimentally demonstrate that an appropriate choice of models contributes to improving the performance with a significant margin.

## 2 BACKGROUND

In this section, we introduce the background which our work is based on.

### 2.1 MACHINE LEARNING FOR COMBINATORIAL OPTIMIZATION

Machine learning approaches for combinatorial optimization problems have been studied in the literature, starting from Hopfield & Tank [17], who applied a variant of neural networks to small instances of Traveling Salesperson Problem (TSP). With the success of deep learning, more and more studies were conducted including Bello et al. [5], Kool et al. [23] for TSP and Wang et al. [37] for MAXSAT.

Khalil et al. [19] proposed an end-to-end reinforcement learning framework S2V-DQN, which attracted attention because of promising results in a wide range of problems over graphs such as MINIMUMVERTEXCOVER and MAXCUT. Another advantage of this method is that it does not require domain knowledge on specific algorithms or any training dataset. It optimizes a deep Q-network (DQN) where the Q-function is approximated by a graph embedding network, called `structure2vec` (S2V) [9]. The DQN is based on their reinforcement learning formulation, where each action is picking up a node and each state represents the "sequence of actions". In each step, a partial solution $S \subset V$, i.e., the current state, is expanded by the selected vertex $v^* = \arg\max_{v \in V(h(S))} Q(h(S), v)$ to $(S, v^*)$, where $h(\cdot)$ is a fixed function determined by the problem that maps a state to a certain graph, so that the selection of $v$ will not violate the problem constraint. For example, in MAXIMUMINDEPENDENTSET, $h(S)$ corresponds to the subgraph of the input graph $G = (V, E)$ induced by $V \setminus (S \cup \mathcal{N}(S))$, where $\mathcal{N}(S)$ is the open neighbors of $S$. The immediate reward is the change in the objective function. The Q-network, i.e, S2V learns a fixed dimensional embedding for each node.

In this work, we mitigate the issue of S2V-DQN's generalization ability. We follow the idea of their reinforcement learning setting, with a different formulation, and replace their Q-learning by a novel learning strategy inspired by AlphaGo Zero. Note that although some studies combine classic heuristic algorithms and learning-based approaches (using dataset) to achieve the state-of-the-art performance [26, 12], we stick to learning without domain knowledge and dataset in the same way as S2V-DQN.

### 2.2 ALPHAGO ZERO

AlphaGo Zero [35] is a well-known superhuman engine designed for use with the game of Go. It trains a deep neural network $f_\theta$ with parameter $\theta$ by reinforcement learning. Given a state (game board), the network outputs $f_\theta(s) = (\boldsymbol{p}, v)$, where $\boldsymbol{p}$ is the probability vector of each move and $v \in [-1, 1]$ is a scalar denoting the state value. If $v$ is close to $1$, the player who takes a corresponding action from state $s$ is very likely to win.

The fundamental idea of AlphaGo Zero is to enhance its own networks by self-play. For this self-play, a special version of Monte Carlo Tree Search (MCTS) [22], which we describe later, is used. The network is trained in such a way that the policy imitates the enhanced policy by MCTS $\boldsymbol{\pi}$, and the value imitates the actual reward from self play $z$ (i.e. $z = 1$ if the player wins and $z = -1$ otherwise).

More formally, it learns to minimize the loss

$$\mathcal{L} = (z - v)^2 + \text{CrossEntropy}(\boldsymbol{p}, \boldsymbol{\pi}) + c_{\text{reg}} \|\theta\|_2^2, \tag{1}$$

where $c_{\text{reg}}$ is a nonnegative constant for $L_2$ regularization.

MCTS is a heuristic search on game trees. In AlphaGo Zero, the search tree is a rooted tree, where each node corresponds to a state and the root is the initial state. Each edge $(s, a)$ denotes action $a$ at state $s$ and stores a tuple $(N(s, a), W(s, a), Q(s, a), P(s, a))$, where $N(s, a)$ is the visit count, $W(s, a)$ and $Q(s, a)$ are the total and mean action value respectively, and $P(s, a)$ is the prior probability. One iteration of MCTS consists of three parts: *select*, *expand*, and *backup*. First, from the root node, we keep choosing an action that maximizes an upper confidence bound

$$Q(s, a) + c_{\text{puct}} P(s, a) \frac{\sqrt{\sum_{a'} N(s, a')}}{1 + N(s, a)}, \tag{2}$$

where $c_{\text{puct}}$ is a non-negative constant (*select*). Once it reaches to unexplored node $s$, then the edge values are initialized using the network prediction $(\boldsymbol{p}, v) = f_\theta(s)$ (*expand*). After expanding a new node, each visited edge is traversed and its edge values are updated (*backup*) so that $Q$ maintain the mean of state evaluations over simulations: $Q(s, a) = \frac{1}{N(s,a)} \sum_{s'|s,a \to s'} v_{s'}$, where the sum is taken over those states reached from $s$ after taking action $a$. After some iterations, the probability vector $\pi$ is calculated by $\boldsymbol{\pi}_a = \frac{N(s_0,a)^{1/\tau}}{\sum_b N(s_0,b)^{1/\tau}}$ for each $a \in A_{s_0}$, where $\tau$ is a temperature parameter.

AlphaGo Zero defeated its previous engine AlphaGo [33] with 100-0 score without human knowledge, i.e., the records of the games of professional players and some known techniques in the history of Go. We are motivated to take advantage of the AlphaGo Zero technique in our problem setting since we also aim at training deep neural networks for discrete optimization without domain knowledge. However, we cannot directly apply AlphaGo Zero, which was designed for two-player games, to combinatorial optimization problems. Section 3.2 and 3.3 explains how we resolve this issue.

### 2.3 GRAPH NEURAL NETWORK

A Graph Neural Network (GNN) is a neural network that takes graphs as input. Kipf & Welling [21] proposed the *Graph Convolutional Network* (GCN) inspired by spectral graph convolutions. Because of its scalability, many variants of spatial based GNN were proposed. Many of them can be described as a Message Passing Neural Network (MPNN) [13]. They recursively aggregate neighboring feature vectors to obtain node embeddings that capture the structural information of the input graph. The *Graph Isomorphism Network* (GIN) [39] is one of the most expressive MPNNs in terms of graph isomorphism. Although they have a good empirical performance, some studies point out the limitation of the representation power of MPNNs [39, 30]. Maron et al. [27] proposed an Invariant Graph Network (IGN) using tensor representations of a graph and was shown to be universal [29, 18]. Since it requires a high-order tensor in middle layers, which is impractical, Maron et al. [28] proposed *2-IGN+*, a scalable and powerful model.

All of these models, as well as S2V [9] used in S2V-DQN, are compared in the experiments to test the difference in the performance for combinatorial optimization. The detail of each model is described in Appendix B.

## 3 METHOD

In this section, we give a detailed explanation of our algorithm to solve combinatorial optimization problems over graphs. First, we introduce our reinforcement learning formulation for NP-hard problems. Then, we explain the basic ideas of our proposed CombOpt Zero in light of the difference between 2-player games and our formulation. Finally, we describe the whole algorithm.

### 3.1 REDUCTION TO MDP

Following S2V-DQN [19], we reduce graph problems into a reinforcement learning setting. Here, we introduce our formulation based on a Markov Decision Process (MDP) [4]. A deterministic MDP is defined as $(S, A_s, T, R)$, where $S$ is a set of states, $A_s$ is a set of actions from the state $s \in S$,

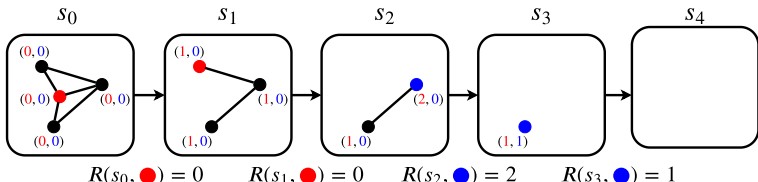

Figure 1: **Example of MaxCut MDP trainsitions.** In the transition sequence, the upper left and center node are colored in ●, the right and lower left node are colored in ●, resulting in the cumulative reward 3.

$T : S \times A_s \to S$ is a deterministic state transition function, and $R : S \times A_s \to \mathbb{R}$ is an immediate reward function. In our problem setting on graphs, each state $s \in S$ is represented as a *labeled* graph, a tuple $s = (G, d)$. $G = (V, E)$ is a graph and $d : V \to L$ is a node-labeling function, where $L$ is a label space.

For each problem, we have a set of terminal states $S_{\text{end}}$. Given a state $s$, we repeat selecting an action $a$ from $A_s$ and transiting to the next state $T(s, a)$, until $s \in S_{\text{end}}$ holds. By this process, we get a sequence of states and actions $[s_0, a_0, s_1, a_1, ..., a_{N-1}, s_N]$ of $N$ steps where $s_N \in S_{\text{end}}$, which we call a trajectory. For this trajectory, we can calculate the simple sum of immediate rewards $\sum_{n=0}^{N-1} R(s_n, a_n)$, and we define $r^*(s)$ be the maximum possible sum of immediate rewards out of all trajectories from state $s$. Let Init be a function that maps the input graph to the initial state. Our goal is to, given graph $G_0$, obtain the maximum sum of rewards $r^*(\text{Init}(G_0))$.

Many combinatorial optimization problems can be accomodated in this framework by appropriately defining $L$, $A_s$, $T$, $R$, Init, and $S_{\text{end}}$ (see Appendix G). Now we take MAXCUT as an example. Let $C \subset E$ be a cut set between $V' \subset V$ and $V \backslash V'$, i.e., $C = \{(u, v) \in E \mid u \in V', v \in V \backslash V'\}$. MAXCUT asks for a subset $V'$ that maximizes the size of cut set $C$. Below, we explain our MDP formulation of MAXCUT.

In each action, we color a node by 0 or 1 and remove it while each node keeps track of how many adjacent nodes have been colored with each color. $A_s = \{(x, c) \mid x \in V, c \in \{0, 1\}\}$ denotes a set of possible coloring of a node, where $(x, c)$ means coloring node $x$ with color $c$. $L = \mathbb{N}^2$, representing the number of colored (and removed) nodes in each color (i.e., $l_0$ is the number of (previously) adjacent nodes of $x$ colored with 0, and same for $l_1$, where $l = d(x)$). Init uses the same graph as $G_0$ and sets $d(x) = (0, 0)$ for all $x \in V$. $T(s, (x, c))$ increases the $c$-th value of $d(x')$ by one for $x' \in N(x)$ and removes $x$ and neighboring edges from the graph. $S_{\text{end}}$ is the states with the empty graphs. $R(s, (x, c))$ is the $(1 - c)$-th (i.e., 1 if $c = 0$ and 0 if $c = 1$) value of $d(x)$, meaning the number of edges in the original graph which has turned out to be included in the cut set (i.e., colors of the two nodes are different).

Figure 1 shows some possible MDP transitions for MAXCUT. It is easy to check that finding a sequence that maximizes the sum of rewards is equivalent to solving the original problem. One of the important differences from the formulation of Khalil et al. [19] is that we do not limit the action space to a set of nodes. This flexibility enables the above formulation, where actions represent *a node coloring*. See Appendix F for the details.

## 3.2 EXTENDING ALPHAGO ZERO

We have seen that many combinatorial optimization problems can be reduced to the deterministic MDP formulation, which is similar to the problem setting of AlphaGo Zero. However, since AlphaGo Zero is designed exclusively for Go (or AlphaZero for two-player games), it still can not be directly applied for our MDP settings. Here, we explain how to extend AlphaGo Zero to our MDP formulation of combinatorial problems with respect to three main differences from Go.

**Graph Input** In our problem setting, the states are represented by (labeled) graphs of different sizes, while the boards of Go can be represented by $19 \times 19$ fixed-size matrix. This can be addressed easily

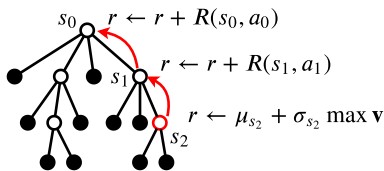

Figure 2: **Backup.** After estimating the reward of the expanded node, we iteratively update the rewards of its ancestors.

by adopting GNN instead of convolutional neural networks in AlphaGo Zero, just like S2V-DQN. The whole state $s = (G, d)$ can be given to GNN, where the coloring $d$ is used as the node feature.

**Range of Final Results** While the final result of Go is only win or lose (or possible draw), the answer to combinatorial problems can take any integer or even real number. AlphaGo Zero models the state value with the range of $[-1, 1]$, meaning that the larger the value is, the more likely the player is to win. A naive extension of this is to directly predict the maximum possible sum of rewards $r^*(s)$. That may, however, grow infinitely in our problem setting usually depending on the graph size, which causes issues. In our preliminary experiments, it did not work well due to difficulty in balancing the scale in equation 1 or in equation 2. For example, in equation 2, the first term could be too large when the answer is large, causing less focus on $P(s, a)$ and $N(s, a)$. To mitigate these issues, we propose a reward normalization technique. Given a state $s$, the network outputs $(\boldsymbol{p}, \boldsymbol{q}) = f_\theta(s)$. While $\boldsymbol{p}$ is the same as AlphaGo Zero (action probabilities), $\boldsymbol{q}$ is a *vector* representing a *normalized* state action value instead of a scalar state value. Intuitively, $\boldsymbol{q}_a$ predicts "how good the reward is compared to the solution obtained by random actions" when taking action $a$ from state $s$. Formally, we train the network so that $\boldsymbol{q}_a$ predicts $(R(s, a) + r^*(T(s, a)) - \mu_s)/\sigma_s$, where $\mu_s$ and $\sigma_s$ are the mean and the standard deviation of the cumulative rewards by random plays from state $s$. Similarly, we also let $W(s, a)$ and $Q(s, a)$ hold the sum and mean of *normalized* action value estimations over the MCTS iterations. By virtue of this normalization, we no longer need to care about the difference of scales in equation 1 and equation 2. When we estimate the state value of $s$, we consider taking action $a$ which maximizes $\boldsymbol{q}_a$ and restoring the unnormalized value by $\mu_s$ and $\sigma_s$:

$$r_{\text{estim}}(s) = \begin{cases} 0 & (s \in S_{\text{end}}), \\ \mu_s + \sigma_s \cdot (\max_{a \in A_s} \boldsymbol{q}_a) & (\text{otherwise}). \end{cases} \tag{3}$$

**Various Immediate Rewards** Since the immediate reward of Go is always zero, the *backup* phase of AlphaGo Zero is simple: Update each edge value so that $Q(s, a)$ keeps the mean of state value predictions over expanded states from $s$ after taking action $a$. Although we have various immediate rewards in our MDP formulations, all of these can be handled easily. Each time we go back to the parent node in the *backup*, we add the immediate reward of the edge to the current reward: $r \leftarrow r + R(s, a)$ when going up path $(s, a)$. Note that since we use normalization technique, $r$ is initialized by equation 3. Figure 2 illustrates the process. We can apply the same strategy when calculating the normalized cumulative reward in data generation (See Section 3.3 for the details).

## 3.3 ALGORITHMS

Based on the discussion so far, now we are ready to introduce the whole algorithm of our proposed CombOpt Zero.

**MCTS** Each edge stores a tuple $(N(s, a), W(s, a), Q(s, a), P(s, a))$ in the same way as AlphaGo Zero. Additionally, each node $s$ stores a tuple $(\mu_s, \sigma_s)$, the mean and the standard deviation of the results by random plays. Given an initial state $s_0$, we repeat the iterations of the three parts: *select*, *expand*, and *backup*. *Select* is same as AlphaGo Zero (keep selecting a child node that maximizes equation 2). Once we reach unexpanded node $s$, we *expand* the node. $(\boldsymbol{p}, \boldsymbol{q}) = f_\theta(s)$ is evaluated and each edge value is updated as $N(s, a) = 0$, $W(s, a) = 0$, $Q(s, a) = 0$, and $P(s, a) = \boldsymbol{p}_a$ for $a \in A_s$. At the same time, $(\mu_s, \sigma_s)$ is estimated by random sampling from $s$. In *backup*, for each node $s'$ and the corresponding action $a'$ in a backward pass, $N(s', a')$ is incremented by one in the same way as AlphaGo Zero. The difference is that $Q(s', a')$ is updated to hold the estimation of the

*normalized* mean reward from $s'$. We approximate the non-normalized state value by equation 3 and calculate the estimated cumulative reward from $s'$ by adding the immediate reward each time we move back to the parent node (Figure 2 illustrates this process). After sufficient iterations, $\boldsymbol{\pi}$ is calculated from $N(s_0, a)$ in the same way as AlphaGo Zero. While the number of the iterations is fixed in AlphaGo Zero, in our proposed CombOpt Zero, since the size of action space differs by states, we make it proportional to the number of actions: $c_{\text{iter}}|A_s|$. Following AlphaGo Zero, we add Dirichlet noise to the prior probabilities only for the initial state $s_0$ to explore the various initial actions. The pseudocode is available in Algorithm 1 in Appendix A.

**Training**  Following AlphaGo Zero, the training of CombOpt Zero is composed of three roles: *data generators*, *learners*, and *model evaluators*. The model that has performed best so far is shared among all of these three components. The *data generator* repeats generating self-play records for a randomly generated graph input, by the MCTS based on the current best model. The records are the sequence of $(s, a, \boldsymbol{\pi}, z')$, which means action $a$ was taken from $s$ depending on the MCTS-enhanced policy $\boldsymbol{\pi}$ and the normalized final cumulative reward was $z'$. $z'$ can be calculated in the same way as *backup* in MCTS: after generating the self-play trajectory, it first calculates the cumulative sum of immediate rewards for each state $s$ in reverse order and normalizes them with $\mu_s$ and $\sigma_s$. The pseudocode of the data generator is available in Algorithm 2 in Appendix A. The *learner* randomly samples mini-batches from the generator's data and updates the parameter of the best model so that it minimizes the loss

$$\mathcal{L} = (z' - \boldsymbol{q}_a)^2 + \text{CrossEntropy}(\boldsymbol{p}, \boldsymbol{\pi}) + c_{\text{reg}}\|\theta\|_2^2. \tag{4}$$

We train $\boldsymbol{q}_a$ to be closer to the normalized cumulative reward $z'$. The *model evaluator* compares the updated model with the best one and stores the better one. Since we can not put two players in a match, the evaluator generates random graph instances each time and compare the performance on them.

## 4 EXPERIMENTS

In this section, we show some brief results of our experiments mainly on MINIMUMVERTEXCOVER, MAXCUT, and MAXIMUMCLIQUE. Refer to Appendix I for the full results and Appendix H for further analyses for each problem.

**Competitors**  Since we aim at solving combinatorial optimization problems without domain knowledge or training dataset, S2V-DQN is our main competitor. Additionally, for each problem, we prepared some known heuristics or approximation algorithms to compare with. For instance, for MINIMUMVERTEXCOVER, we implemented a simple randomized 2-approximation algorithm (2-approx) and an interger programming by CPLEX solver. For MAXCUT, we used a randomized algorithm by semidefinite programming [14] and two heuristics, non-linear optimization with local search [6] and cross-entropy method with local search [24] from MQLib [10]. See Appendix H for competitors of other problems and the details.

**Training and Test**  We trained the models for two hours to make the training of both CombOpt Zero and S2V-DQN converge (Only 2-IGN+ was trained for four hours due to its slow inference). Note that CombOpt Zero was trained on 4 GPUs while S2V-DQN uses a single GPU because of its implementation, which we discuss in Section 4.4. For the hyperparameters of the MCTS, we referred to the original AlphaGo Zero and its reimplementation, ELF OpenGo [36]. See Appendix C for the detailed environment and hyperparameters. Since we sometimes observed extremely poor performance for a few models both in S2V-DQN and CombOpt Zero when applied to large graphs, we trained five different models from random parameters and took the best value among them at the test time. Throughout the experiments, all models were trained on randomly genrated Erdős-Renyi (ER) [11] graphs with $80 \leq n \leq 100$ nodes and edge probability $p = 0.15$ except for MAXCUT, where nodes were $40 \leq n \leq 50$ and for MAXIMUMCLIQUE, where edge probability was $p = 0.5$. As the input node feature of MAXCUT, we used a two-dimensional feature vector that stores the number of adjacent nodes of color 1 and color 2. For the other problems, we used a vector of ones. In tests, to keep the fairness, CombOpt Zero conducted a greedy selection on network policy output $\boldsymbol{p}$, which is the same way as S2V-DQN works (except for Section 4.2 where we compared the greedy selection and MCTS).

Table 1: **Generalization Comparision between CombOpt Zero and S2V-DQN.** COZ and DQN are short for CombOpt Zero and S2V-DQN respectively. Smaller is better for MINIMUMVERTEX-COVER and larger is better for MAXCUT. Three sections stand for Erdős-Renyi models, other synthric graphs such as BA graphs and Watts-Strogatz graphs, and real-world graphs.

| | MVC | | MAXCUT | | | MVC | | MAXCUT | |
|---|---|---|---|---|---|---|---|---|---|
| | COZ | DQN | COZ | DQN | | COZ | DQN | COZ | DQN |
| er100_15 | **76** | **76** | 494 | **527** | | | | | |
| er1000_5 | 900 | **898** | 12561 | **14424** | | | | | |
| er5000_1 | 4484 | **4482** | 63389 | **71909** | | | | | |
| ba100_5 | **63** | **63** | 337 | **343** | cora | **1258** | **1258** | **4260** | 4243 |
| ba1000_5 | **592** | 594 | **3492** | 3465 | citeseer | 1462 | **1461** | **3933** | 3893 |
| ba5000_5 | **2920** | 2927 | **17381** | 16870 | web-edu | **1451** | **1451** | **4712** | 4289 |
| ws100_2 | **49** | **49** | **98** | 97 | web-spam | **2299** | 2319 | 20645 | **21027** |
| ws100_10 | **78** | **78** | **335** | **335** | road-min... | **1324** | 1329 | **3080** | 3015 |
| ws1000_2 | **492** | 496 | **999** | 973 | bio-yeast | **456** | 457 | **1769** | 1751 |
| ws1000_4 | **635** | 636 | **1536** | 1327 | bio-SC-... | **1039** | 1053 | 10893 | **11890** |
| ws1000_10 | 787 | **784** | **3312** | 3287 | rt_dama... | **369** | **369** | **3698** | 3667 |
| reg_100_5 | **63** | 64 | **207** | 205 | soc-wiki... | 407 | **406** | **2119** | 2064 |
| reg_1000_5 | **632** | 634 | **2051** | 2046 | socfb-bo... | 1796 | **1793** | **42063** | 37140 |
| tree100 | **44** | **44** | **99** | 98 | | | | | |
| tree1000 | **439** | 440 | **999** | 982 | | | | | |

Table 2: **Improvement by test-time MCTS for** MAXIMUMCLIQUE. Larger is better. Results with test-time MCTS are shown in the parentheses. Bold values means the best solution among learning based approaches (COZ and S2V-DQN). Underlined values are the best among all methods.

| | CombOpt Zero | | | | | | | | S2V-DQN | CPLEX |
|---|---|---|---|---|---|---|---|---|---|---|
| | 2-IGN+ | | GIN | | GCN | | S2V | | | |
| cora | 4 | (**5**) | 4 | (**5**) | 3 | (**5**) | 4 | (**5**) | 4 | 5 |
| citeseer | 4 | (**6**) | 5 | (**6**) | 4 | (**6**) | 4 | (**6**) | 4 | 6 |
| web-edu | 16 | (**30**) | 16 | (**30**) | 16 | (16) | 16 | (**30**) | 16 | 30 |
| web-spam | 10 | (**20**) | 16 | (17) | 7 | (17) | 16 | (17) | 16 | 20 |
| soc-wiki-vote | 5 | (**7**) | 6 | (**7**) | 6 | (**7**) | 6 | (**7**) | 6 | 7 |
| socfb-bowdoin47 | 14 | (**23**) | 15 | (**23**) | 7 | (22) | 13 | (**23**) | 14 | 23 |

**Dataset** We generated ER and Barabási-Albert (BA) graphs [3] of different sizes for testing. ER100_15 denotes an ER graph with 100 nodes and edge probability $p = 0.15$ and BA100_5 denotes a BA graph with 100 nodes and 5 edges addition per node. Also, we used 10 real-world graphs from Network Repository [32], including citation networks, web graphs, bio graphs, and road map graphs, all of which were handled as unlabeled and undirected graphs. Refer to Appendix H for the full results on these 10 graphs. For MINIMUMVERTEXCOVER and MAXIMUMINDEPENDENTSET, we additionally tested on DIMACS [2], difficult artificial instances. We generated other synthetic instances in Section 4.1.

## 4.1 COMPARISON OF GENERALIZATION ABILITY

We compared the generalization ability of CombOpt Zero and S2V-DQN to various kinds of graphs. To see the pure contribution of CombOpt Zero, here CombOpt Zero incorporated the same graph representation model as S2V-DQN, namely S2V. Table 1 shows the performance for MINIMUMVER-TEXCOVER and MAXCUT on various graph instances (see Appendix C for the explanation of each graph). While S2V-DQN had a better performance on ER graphs, which were used for training, CombOpt Zero showed a better generalization ability to the other synthetic graphs such as BA graphs, Watts-Strogatz graphs [38], and regular graphs (i.e., graphs with the same degree of nodes). It was interesting that CombOpt Zero successfully learned the optimal solution of MAXCUT on trees (two-coloring of a tree puts all the edges into the cut set), while S2V-DQN does not. Appendix E

---

[2] https://turing.cs.hbg.psu.edu/txn131/vertex_cover.html

visualizes how CombOpt Zero achieves the optimal solution on trees. CombOpt Zero also generalized better to real-world graphs although S2V-DQN performed better on a few instances, which might be because some real-world graphs have similar characteristics as ER random graphs.

## 4.2 Test-time MCTS

The greedy selection in Sections 4.1 and 4.3 does not make full use of CombOpt Zero. When more computational time is allowed in test-time, CombOpt Zero can explore better solutions using the MCTS. Since the MCTS on large graphs takes a long time, we chose MaximumClique as a case study because solution sizes and the MCTS depths are much smaller than the other problems. We used the same algorithm as in the training (Algorithm 1) with the same iteration coefficient ($c_{iter} = 4$) and $\tau = 0$, and selected an action based on the enhanced policy $\pi$. In all the instances, the MCTS finished in a few minutes on a single process and a single GPU, thanks to the small depth of the MCTS tree. The improvements are shown in Table 2. Note that the solutions by CPLEX except for scfb-bowdoin was optimal since the execution finished within the cut-off time. Considering this fact, CombOpt Zero with test-time MCTS gaining the same result as CPLEX on all the instances, while S2V-DQN on no instances, is promising.

## 4.3 Combination with Different GNN Models

We compared the performance of CombOpt Zero with four different GNN models (2-IGN+, GIN, GCN, and S2V). See Appendix H for all the results of the 5 problems. One interesting insight was that the best GNN models were different across the problems. For example, while GIN had the best performance in MaxCut, GCN performed slightly better in MinimumVertexCover. Although GCN performed significantly worse in MaxCut, it did almost the best for the other four NP-hard problems. 2-IGN+, a theoretically more expressive GNN, did not work well except for MaximumClique. Since we observed that the performance on training instances did not significantly differ among different GNN models, the test performance difference is possibly because of the generalization ability of each GNN for specific combinatorial structure, which we put as future research. Overall, CombOpt Zero was strongly enhanced with a proper GNN model. Also, it is worth noting that, in a few instances, CombOpt Zero outperformed CPLEX on MinimumVertexCover or SOTA heuristic solvers on MaxCut (see Table 5 and Table 6 in Appendix I).

## 4.4 Tradeoffs between CombOpt Zero and S2V-DQN

Here, we summarize some characteristics of CombOpt Zero and S2V-DQN. During the training, CombOpt Zero used 32 processes and four GPUs as described in Appendix C, while S2V-DQN used a single process and GPU because of its implementation. CombOpt Zero takes a longer time to generate self-play data than S2V-DQN due to the MCTS process. For this reason, CombOpt Zero needs a more powerful environment to obtain stable training. On the other hand, since CombOpt Zero is much more sample-efficient than S2V-DQN (see Appendix D), the bottleneck of the CombOpt Zero training is the data generation by the MCTS. This means that it can be highly optimized with an enormous GPU or TPU environment as in Silver et al. [35], Silver et al. [34], and Tian et al. [36].

By its nature, S2V-DQN can be also combined with other GNN models than S2V. However, S2V-DQN is directly implemented with GPU programming, it is practically laborious to combine various GNN models. On the other hand, CombOpt Zero is based on PyTorch framework [31] and it is relatively easy to implement different GNN models.

## 5 Conclusion

In this paper, we presented a general framework, CombOpt Zero, to solve combinatorial optimization on graphs without domain knowledge. We showed that CombOpt Zero worked well for combinatorial problems in relatively small computational resources (four GPUs) compared to the original AlphaGo Zero. The Monte Carlo Tree Search (MCTS) in the training time successfully helped the wider exploration than the existing method and enhanced the generalization ability to various graphs. Another advantage of our method is that the test-time MCTS significantly strengthened the performance if more computation time is allowed.

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
