# OpenReview forum: "Solving NP-Hard Problems on Graphs with Extended AlphaGo Zero"
_ICLR.cc/2021/Conference — Reject_

### Official Review · AnonReviewer2 · 2020-10-15
**Good idea, but experimental results too weak to conclude anything**

**Rating:** 4
**Confidence:** 4

**Review:**

There has been a sequence of recent works on learning heuristics for combinatorial optimization problems on graphs by treating them as Markov decision processes, and learning by reinforcement a good policy. Since the dynamics of these problems can be readily simulated, in this paper, the authors propose to use AlphaZero, a MCTS variant, with a GNN architecture to learn and predict good solutions. They compare against the approach of Khalil et al. (2017), as well as hand-made heuristics, on three benchmark problems (Min Vertex Cover, Max Cut, Max Clique). There are results on Max Independent Set and Min Feedback Vertex Set in the appendix as well.

I think the main contribution of the paper is in thinking to apply MCTS/AlphaZero to these graph-based combinatorial optimization problems. To do this they need to modify the algorithm a bit, albeit in relatively straightforward ways. Now, since there is no theory, everything relies on the strength of the experimental results, but on this aspect I have issues.

First, the authors seem to disregard relevant work done after Khalil et al. (2017). At minimum, I think comparisons with Li et al. (2018) and Mittal et al. (2019) would be warranted. Both of these papers compare against Khalil et al.'s S2V-DQN, and predate this paper, so I think comparisons against them are a must. In addition, relevant concurrent work includes Karalias and Loukas (2020) and Drori et al. (2020), which could be interesting baselines to compare. Also, in terms of literature review, it would be relevant to discuss the recent article of Xing et al. (2020) which also does MCTS, albeit for the TSP only. More generally, there is a large amount of work done with focus on the TSP/VRP, some of which could probably be generalized to any graph-based problem without too much trouble, which should at least be discussed. So the literature review is incomplete and the experimental section is missing competitors.

In addition, I found the paper not very well written. In addition to many typos, some details, even after digging in the supplementary materials, are left unclear to me.
1) On p. 8, you write that "[a]lso, it is worth noting that, in a few instances, CombOpt Zero outperformed CPLEX on MinimumVertexCover", which seems to indicate that there is a time limit? If so, what is it? The supplementary materials mention (p. 21) that "[e]mpty cells mean execution did not finish within 2 hours.", which suggest that there's a 2h time limit for resolution? Or does this refer to the training time, since you write on p. 6 that "[e]e trained the models for two hours to make the training of both CombOptZero and S2V-DQN converge"? And if there is a 2h time limit for all the methods, why do you seem to limit heuristics to 10 min, as indicated on p. 14: "[w]e set the time limit of 10 minutes for these algorithms and used the best found solution as the results."? Why not give them 2 hours like the rest?
2) All these problems have integer programming formulations, so why compare against CPLEX only for Min Vertex Cover? Why not have it as competitor for Max Cut and Max Clique as well?
3) Why is there no timings? How long did it take for the heuristics to find the solutions at test time, compared to the AlphaZero model? Both for the greedy and MCTS decoding. This should be done despite the difference in hardware.
4) I think test-time MCTS comparisons should be reported for all problem classes, not just MaximumClique.

Finally, the improvements themselves don't seem particularly big, at least in the greedy decoding case. There's only a few instance classes (all for MaxCut) where there seems to be any significant gain, and without reporting inference time, I'm not sure we can conclude much. As for the MCTS decoding, is there a way of running a similar algorithm from the learned S2V-DQN? Also, the results don't seem to have been repeated over several seeds, which should be done. This is on top of the fact that you use the best prediction among 5 different models - that is, the whole process should be repeated many times, with average objective value and test inference times reported, along with the standard deviations (see Henderson et al. 2017). For example, for the first run, the five models could use seeds 0 to 4; then for the second run, the five models could use seeds 5 to 9; etc. You could then report the average and standard deviation over the total number of runs.

Overall, I think the approach that this paper proposes, namely using a MCTS/AlphaZero-type approach to learn heuristics for graph-based combinatorial optimization problems, has good potential to improve on pure RL methods. After all, since the dynamics are simple enough that we can simulate, it makes sense to make use of that fact. However, at least with the current setup, the experimental section appears too weak for me to be able to conclude that there is really an improvement, with timings, baselines and multiple seeds missing. Moreover, even with a stronger experimental methodology, right now it is not clear to me if the gains are significant. Based on this assessment I would recommend rejection.

References

[1] Li, Z., Chen, Q., & Koltun, V. (2018). Combinatorial optimization with graph convolutional networks and guided tree search. In Advances in Neural Information Processing Systems (pp. 539-548).

[2] Mittal, A., Dhawan, A., Manchanda, S., Medya, S., Ranu, S., & Singh, A. (2019). Learning heuristics over large graphs via deep reinforcement learning. arXiv preprint arXiv:1903.03332.

[3] Xing, Z., Tu, S., & Xu, L. (2020). Solve Traveling Salesman Problem by Monte Carlo Tree Search and Deep Neural Network. arXiv preprint arXiv:2005.06879.

[4] Karalias, N., & Loukas, A. (2020). Erdos Goes Neural: an Unsupervised Learning Framework for Combinatorial Optimization on Graphs. arXiv preprint arXiv:2006.10643.

[6] Drori, I., Kharkar, A., Sickinger, W. R., Kates, B., Ma, Q., Ge, S., ... & Udell, M. (2020). Learning to Solve Combinatorial Optimization Problems on Real-World Graphs in Linear Time. arXiv preprint arXiv:2006.03750.

[7] Henderson, P., Islam, R., Bachman, P., Pineau, J., Precup, D., & Meger, D. (2017). Deep reinforcement learning that matters. arXiv preprint arXiv:1709.06560.

---

> ### Author Response · Authors · 2020-11-24
> **Thank you for the comments!**
>
> Thank you for the detailed feedback. Here, we address your questions.
>
> Q: Time limit of the test
> A: We set a time limit for neural methods 2 hours. This is almost for IGN+, which requires $\mathrm{\Theta}(V^3)$ of time complexity for inference. Although our implementation of other GNN models takes about 40 minutes for some large sparse instances due to dense matrix multiplication, considering the execution of S2V-DQN, which is based on sparse matrix multiplication, finishes in a few minutes for all the instances, we think it is still reasonable to set a 10-minute time limit for heuristics.
>
> Q: All these problems have integer programming formulations, so why compare against CPLEX only for Min Vertex Cover?
> A: We compared our method against CPLEX for Maximum Clique and Maximum Independent Set as well as Minimum Vertex Cover.
>
> Q: Why is test-time MCTS only done for MaxClique?
> A: The test-time MCTS requires a long time if the search tree is deep. During training, we trained the network with small random graphs, so MCTS runs quickly, but in the testing phase, some input graphs have thousands of nodes and it’s almost impossible to do MCTS on it. In MaxClique, since the solution size is small, it finishes quickly.
>
> Also, we appreciate your feedback on our experimental methodologies, such as timings and random seeds, which we will reflect on the future revision.

---

### Official Review · AnonReviewer1 · 2020-10-24
**Method makes sense, needs more info on empirical evals**

**Rating:** 5
**Confidence:** 4

**Review:**

This paper proposes an AlphaGo Zero style algorithm for training policies for solving combinatorial optimization problems. The main idea is to generate training data with MCTS for both a policy network and a value network via self-play. Different graph neural networks (GNNs) are considered as learning models to compare their performances. Empirical evaluations on 5 NP-hard class of problems are provided to demonstrate better performance than an existing RL method S2V-DQN. Comparisons are also provided with CPlex and other approximation algorithms for specific classes of problems.

Positives:
1. The adaptation of the AlphaGo Zero algorithm to combinatorial optimization problem is novel and natural. Some detailed decisions, such as normalizing reward vectors by the distribution generated with random actions, demonstrate careful considerations, and are likely to provide insights for subsequent work on adaptations for other problems.

2. The comprehensive evaluations for different GNN architectures are welcome. As one would expect, there are variations in which architecture is the best for different classes of problems.

Negatives:
1. The empirical evaluations could be improved with the following additions:
a) Test-time performance of MCTS with S2V-DQN in Table 2
b) Error bars for the results, as well as information on how many instances per problem class are used for training and testing.
c) wall-clock time comparisons: using MCTS is used in test-time, how long would the actual wall-clock take?

2. The writing could be improved a lot. It is very difficult to evaluate the paper without reading the Appendix, which contains lots of important details on the algorithm and models. For example,
a) Algorithm 2 is key in describing the data generation procedure and I found it was much easier to understand it by reading both the text description as well as the pseudocode. I suggest including it in the main paper.
b) The main paper didin’t mention that the MCTS results were shown with S2V as the GNN. This information is in Appendix D. It is important to know that the comparison with S2V-DQN is fair.
c) Lots of experimental results are in the Appendix as well. I understand there is a page limit but including and discussing them in the main paper would make the paper more convincing.

3. The description for the model f_\theta is incomplete. While the Appendix describes various GNNs up to the feature extraction step, the information on how the mapping from features to final predictions, action distribution and value estimations, is missing. Please provide them in the

========
Post-rebuttal comments:
Thank the authors for answering my questions. I think the current version of the paper is below the bar of acceptance at ICLR and I hope the authors can incorporate the answers to make the submission stronger in the future.

---

> ### Author Response · Authors · 2020-11-24
> **Thank you for the comments!**
>
> We appreciate your insightful comments and feedback. Here are the answers to your questions.
>
> Q: Test-time performance of MCTS with S2V-DQN in Table 2
> A: We cannot run MCTS by S2V-DQN because MCTS of AlphaGo Zero requires both value network and policy network, while S2V-DQN trains only Q-function.
>
> Q: information on how many instances per problem class are used for training and testing
> A: Figure 3 and 4 in Appendix show the number of instances used for training.
>
> Q: using MCTS is used in test-time, how long would the actual wall-clock take
> A: As written in Section 4.2, test-time MCTS finished in a few minutes on a single process and a single GPU on all instances.
>
> Q: The description for the model f_\theta is incomplete.
> A: We apologize that we did not include this information in the paper. Since our shape of action distribution and value estimation are the same, we simply output $R^{2 \times n}$ array (We apply softmax function for action distribution and normalization for value estimation).
>
> Also, thank you for pointing out some shortcomings in writing, which we will reflect in the final version of the paper.

---

### Official Review · AnonReviewer3 · 2020-10-29
**Improved performance, but difficult to say what was learned**

**Rating:** 4
**Confidence:** 4

**Review:**

This paper applies AlphaGo Zero to solve combinatorial optimization problems on graphs, replacing the CNN with a graph neural network. The results show that the system typically generalizes better than previous work by Khalil et al to new graph distributions.

On the positive side, developing techniques which better generalize across problem distributions is an important area in which to improve learning-to-optimize techniques. The experimental results do seem to show that CombOpt Zero has an edge over the results from Khalil et al, often matching the CPLEX solution when test-time MCTS is allowed (though perhaps this shouldn't be counted too heavily, since typically the motivation for such techniques is fast test-time performance).

However, I do not think that the paper makes a sufficient contribution to warrant publication at ICLR. The techniques used are mostly an off-the-shelf application of AlphaGo Zero, with the modifications (swapping a GNN in instead of the CNN and normalizing the rewards) being fairly direct. Investigations of off-the-shelf methods can of course be valuable, but I don't think that enough insight was gained from the experimental results in this case. Why is the AlphaGo Zero architecture/training method more effective here? What about it leads to more effective generalization, even though the in-distribution performance is roughly equal? Ultimately, it would be ideal if there was something to be learned from the results beyond the fact that this method offers an empirial advantage in some situations.

---

> ### Author Response · Authors · 2020-11-24
> **Thank you for the comments!**
>
> We thank you for the helpful comments. We think that one of the reasons why CombOpt Zero generalizes better than S2V-DQN is because MCTS explores a wider range of actions than the Q-learning strategy. We leave the theoretical analysis as future work. Also, Appendix E visualizes and considers some insights from our experiments.
>
> Also, we would like to remind you of some other contributions of our paper. First, one of the main contributions over S2V-DQN is that our method can exploit MCTS in evaluation, which much improves existing tree search [1]. Especially for MaxClique, there is a significant difference in the solution size between CombOpt Zero and S2V-DQN. Secondly, there are many studies on the application of machine learning for combinatorial problems, but few of them are focused on classical problems where sophisticated approximation algorithms or heuristics are developed. Our paper is one of the early papers that extensively compared recent GNNs with such heuristics. And we believe that our study opens up a new direction for applying machine learning to classical combinatorial problems.
>
> [1] Li, Z., Chen, Q., & Koltun, V. (2018). Combinatorial optimization with graph convolutional networks and guided tree search. In Advances in Neural Information Processing Systems

---

### Decision · Program_Chairs · 2021-01-07
**Final Decision**

**Decision:**

Reject

**Comment:**

This paper proposes applying AlphaGo Zero style ideas for solving combinatorial optimization problems over graphs with two changes:
Replacing CNNs with graph neural networks
Normalization of rewards

The reviewers raised valid points about this paper.
1. Lack of technical novelty
2. Experimental results are not strong enough to draw meaningful conclusions
3. Since techniques are mostly off-the-shelf, extracting general knowledge, insights and conclusions from empirical evaluation is important, but unfortunately was missing.

I agree with the review comments. Overall, my assessment is that the paper requires more work  before it is ready for publication.